# PPARα Agonist Suppresses Inflammation after Corneal Alkali Burn by Suppressing Proinflammatory Cytokines, MCP-1, and Nuclear Translocation of NF-κB

**DOI:** 10.3390/molecules24010114

**Published:** 2018-12-29

**Authors:** Yuichiro Nakano, Masaaki Uchiyama, Takeshi Arima, Shinya Nagasaka, Tsutomu Igarashi, Akira Shimizu, Hiroshi Takahashi

**Affiliations:** 1Department of Ophthalmology, Nippon Medical School, Tokyo 113-8603, Japan; nakano1212@nms.ac.jp (Y.N.); uchiyama@nms.ac.jp (M.U.); takesuiii0714@nms.ac.jp (T.A.); tutomu@nms.ac.jp (T.I.); tash@nms.ac.jp (H.T.); 2Department of Analytic Human Pathology, Nippon Medical School, Tokyo 113-8602, Japan; 3mousquetaires51a@gmail.com

**Keywords:** fenofibrate, IκB-α, IL-1β, ophthalmic solution, VEGF-A

## Abstract

We investigated the effect of a peroxisome proliferator-activated receptor α (PPARα) agonist after corneal alkali injury. Fenofibrate 0.05% (PPARα agonist group) or vehicle (Vehicle group) was topically instilled onto the rat cornea after injury. Histological, immunohistochemical, and real-time reverse transcription PCR analyses were performed. PPARα-positive cells were observed among basal cells of the corneal epithelium in normal and alkali-burned corneas. The number of infiltrating neutrophils and macrophages at the corneal limbus was lower in the PPARα agonist group. Interleukin-1β (IL-1β), IL-6, IL-8, monocyte chemoattractant protein-1 (MCP-1), and vascular endothelial growth factor-An mRNA expression was suppressed in the PPARα agonist group compared to the Vehicle group. mRNA levels of nuclear factor kappa B (NF-κB) in corneal tissue were not different. However, NF-κB was expressed in the cytoplasm of basal cells in the PPARα agonist group and in the nucleus in the Vehicle group. MCP-1 was more weakly expressed in the PPARα agonist group. The PPARα agonist inhibited inflammation during the early phase after injury. Anti-inflammatory effects of the PPARα agonist included prevention of up-regulation of proinflammatory cytokines and MCP-1, and prevention of inflammatory cell infiltration into the injured cornea. Thus, a PPARα agonist may be a promising treatment for corneal injury.

## 1. Introduction

For good vision, transparency of the cornea is essential. Injuries that damage the corneal stroma often result in scarring. Since inflammation and angiogenesis are deeply involved in scar tissue formation, agents that can suppress these phenomena have long been sought. One of the new suppression candidates is peroxisome proliferator-activated receptors (PPARs). PPAR is a nuclear receptor that belongs to the steroid hormone receptor superfamily. Three isoforms of PPARs, which vary in their tissue distribution, have been confirmed and include PPARα, β, and γ [1,2,3]. PPARα is localized in the liver, kidney, muscle, and heart; β is ubiquitously expressed in many organs and tissues; and γ is present in adipocytes and the small intestine [4,5]. Drugs that modulate PPARs are commonly used in clinical practice today. For example, selective PPARα agonists such as fenofibrate are routinely used for the treatment of hyperlipidemia, whereas selective PPARγ agonists such as thiazolidinediones are used for the treatment of diabetes [6]. Selective PPARβ agonists have also been examined regarding their usefulness in treating metabolic syndrome [7]. In addition to their role in the transcriptional regulation of metabolism, PPARs may be involved in controlling various inflammatory, angiogenic, and fibrotic physiological processes [8]. Thus, PPARs may represent novel targets for the treatment of inflammatory and vascular diseases [9].

Although the localization of PPARs in the cornea has not been well demonstrated, some studies have revealed that PPARs play a role in pathological conditions in this tissue. PPARβ is upregulated during corneal epithelial wound healing, and pretreatment with a PPARβ agonist inhibits inflammation-induced corneal epithelial cell death [10]. We also previously reported that an ophthalmic solution of a PPARγ agonist inhibits inflammation, decreases the fibrotic reaction, and prevents neovascularization in the cornea after alkali burn injury [11]. In addition, we reported that a PPARα agonist suppresses neovascularization by reducing vascular endothelial growth factor (VEGF) and angiopoietin-2 in the cornea after alkali burn injury [12]. A recent study also demonstrated protective and anti-inflammatory effects of a PPARα agonist in an experimental uveitis model [13].

PPARα agonists may regulate inflammation by maintaining a negative regulator, nuclear expression of the kappa light polypeptide gene enhancer in the B cell inhibitor, alpha (IκB-α). Nuclear factor kappa B (NF-κB) is a major transcription factor that plays a role in initiating proinflammatory immune responses. IκB-α mediates anti-inflammatory effects by inhibiting NF-κB activation. PPARα agonists exert anti-inflammatory activities in different cell types by antagonizing the transcriptional activity of NF-κB and inducing the expression of IκB-α [14,15]. However, the role of PPARα in corneal inflammation has yet to be definitively established.

In the present study, we investigated the localization of PPARα in rat cornea and explored the effect of an ophthalmic solution of the selective PPARα agonist, fenofibrate, following corneal alkali burn.

## 2. Results

### 2.1. Localization of PPARs in the Rat Cornea

In the normal rat cornea, PPARα-, β-, and γ-positive cells were observed among the epithelial basement cells (Figure 1A–C).

A corneal burn was created in rats by treating the cornea with NaOH. At day 7 in the cornea after alkali burn, PPARα, β, and γ were expressed in the re-epithelialized basement cells (Figure 1D–F). In the epithelial basement cells in the center at 6 h and the limbus at day 1, more PPARα-positive cells (Figure 2A,B) were found than PPARβ- (Figure 2C,D) and γ-positive cells (Figure 2E,F). Furthermore, PPARα, β, and γ were expressed on infiltrating neutrophils and macrophages at day 7 (Figure 3).

### 2.2. Effects of an Ophthalmic Solution of a PPARα Agonist after Alkali Burn Injury

To test the effects of the PPARα agonist, fenofibrate, on the cornea after alkali burn injury, an ophthalmic solution of fenofibrate (PPARα agonist group) or vehicle (Vehicle group) was topically instilled onto the rat’s ocular surfaces after the alkali injury. The number of infiltrating inflammatory cells, such as neutrophils and macrophages, was significantly lower in the PPARα agonist group versus the Vehicle group at 6 h and on day 2 (Figure 4 and Figure 5). The number of cells observed was: neutrophils; 6 h: 46.5 ± 11.47 cells/×400 high-power field (HPF) in the Vehicle group, 30.44 ± 6.11 cells/HPF in the PPARα agonist group, *p* < 0.01; day 1: 45.6 ± 20.43 cells/HPF in the Vehicle group, 37.21 ± 20.21 cells/HPF in the PPARα agonist group; day 2: 53.07 ± 16.56 cells/HPF in the Vehicle group, 23.8 ± 12.8 cells/HPF in the PPARα agonist group, *p* < 0.01. Macrophages; 6 h: 39.0 ± 14.47 cells/HPF in the Vehicle group, 22.88 ± 13.16 cells/HPF in the PPARα agonist group, *p* < 0.01; day 1: 49.5 ± 19.93 cells/HPF in the Vehicle group, 34.36 ± 16.35 cells/HPF in the PPARα agonist group; day 2: 54.43 ± 16.7 cells/HPF in the Vehicle group, 31.5 ± 9.05 cells/HPF in the PPARα agonist group, *p* < 0.01.

Real-time RT–PCR was performed to assess the expression of genes encoding proinflammatory cytokines (interleukin [IL]-1β and IL-6), chemotactic chemokines (IL-8 and monocyte chemoattractant protein-1 (MCP-1)), and a factor involved in neovascularization (VEGF-A). In the alkali-burned corneas, the levels of mRNA for the proinflammatory chemokines and cytokines, IL-1β, IL-6, IL-8, MCP-1, and VEGF-A, were increased at 6 h during development of corneal inflammation in the Vehicle and PPARα agonist groups (Figure 6A–E). However, the increases in these molecules were suppressed by treatment with the ophthalmic solution of the PPARα agonist after alkali exposure starting from 6 h. The group treated with the ophthalmic solution of the 0.05% PPARα agonist exhibited suppressed levels of IL-1β, IL-6, and IL-8. We also observed an increase in the level of PPARα in the PPARα agonist group versus the Vehicle group (Figure 6F). For the alkali-burned corneas, we found no significant differences in the levels of NF-κB or IκB-α at 6 h during the development of corneal inflammation in the PPARα agonist or Vehicle groups (Figure 6G,H).

### 2.3. NF-κB, IκB-α, and MCP-1 Staining on Day 1

In the normal rat cornea, NF-κB-positive areas showed weak, diffuse labeling of the cytoplasm of the epithelial basement cells in the center (Figure 7A) and limbus (Figure 7D). The epithelial basement cells displayed prominent nuclear NF-κB labeling throughout the epithelial basement cells during re-epithelialization, with weaker diffuse labeling of the cytoplasm observed in the alkali-burned cornea in the center (Figure 7B) and limbus (Figure 7E) in the Vehicle group. In the alkali-burned corneas, NF-κB-positive areas were observed in the cytoplasm of the epithelial basement cells in the center (Figure 7C) and limbus (Figure 7F) in the PPARα agonist group.

In the normal cornea, no apparent labeling of IκB-α in the cytoplasm of the epithelial basement cells in the center (Figure 8A) and limbus (Figure 8D). In the alkali-burned cornea, IκB-α-positive labeling was stronger in the PPARα agonist group in the central cornea (Figure 8C) and limbus (Figure 8F) compared to these regions of the Vehicle group (Figure 8B,E). Diffuse labeling of the cytoplasm of the epithelial basement cells was observed during re-epithelialization.

The expression of MCP-1 were not observed in the center (Figure 9A) or limbus (Figure 9D) of the normal rat cornea. In the alkali-burned cornea, diffuse expression of the epithelial basement cells was observed during re-epithelialization in Vehicle group (Figure 9B,E). However, the expression of MCP-1 was weaker in the center (Figure 9C) and limbus (Figure 9F) of the PPARα agonist group compared to these regions of the Vehicle group.

## 3. Discussion

It has been reported that PPARα, β, and γ plays different roles in corneal wound healing. Administration of PPARα not only reduces inflammation but also corneal neovascularization [12]. PPARβ is involved in tissue repair and administration of PPARβ agonist promotes the healing of corneal epithelial wounds [10]. The ophthalmic solution of the PPARγ agonist inhibits inflammation and furthermore promotes corneal wound healing by activating M2 macrophage [11]. Although localization of PPARs has been reported in various organs, localization in the cornea has yet to be shown. In our present study, we demonstrated the localization of PPARs in the rat cornea. PPARα-, β-, and γ-positive cells were observed among the basal cells of the epithelium in the normal cornea and in re-epithelialized basal cells after alkali burn. In addition, PPARα, β, and γ were expressed in the infiltrating neutrophils and macrophages. Among the observed subtypes, PPARα-positive cells were primarily found in the corneal regenerating epithelium, limbal epithelium, and infiltrating inflammatory cells. These results strongly suggest that PPARα plays an important role in the wound healing process after a corneal burn.

Correspondingly, immunohistochemical and real-time RT–PCR analyses revealed that the PPARα agonist, fenofibrate, inhibited proinflammatory reactions by decreasing the expression of IL-1β, IL-6, IL-8, and MCP-1. These findings are in agreement with previous studies that showed that a PPARα agonist plays a role in regulating the inflammatory response [14,16,17,18]. In addition, fenofibrate decreased the expression of VEGF-A, which may be associated with suppressing neovascularization during the early phase after an alkali injury [12]. In addition, in our current study, the PPARα agonist, fenofibrate, decreased the expression of VEGF-A, which may be associated with suppressing neovascularization during the early phase after an alkali injury. The results were in good accordance with previous studies [12,19]. Furthermore, immunohistochemical and real-time RT–PCR analyses demonstrated that fenofibrate also increased the expression of PPARα, which suggests self-inducing effects.

We then investigated the effect of the PPARα agonist on the expression of NF-κB, which is a ubiquitous transcription factor that regulates key processes such as inflammation, apoptosis, stress response, wound healing, angiogenesis, and lymphangiogenesis [20,21,22]. Pro-inflammatory cytokines, IL-1β, IL-6, IL-8, chemokine MCP-1, and VEGF are deeply involved in inflammation. In the process, activated NF-κB is known to be associated with production of pro-inflammatory cytokines, IL1β, IL6, and IL8. In unstimulated cells, NF-κB is sequestered in the cytoplasm by IκB-α, which is a suppressor protein that binds to NF-κB. NF-κB activation is initiated by a signal that induces IκB degradation. NF-κB then enters the nucleus and activates specific genes [23,24,25]. In the present study, our real-time RT–PCR analysis did not identify significantly different levels of NF-κB or IκB-α between the PPARα agonist and Vehicle groups. However, our immunohistochemical analysis demonstrated the effects of the PPARα agonist on NF-κB expression. NF-κB was expressed in the cytoplasm of the basal cells in the PPARα agonist group, whereas this protein was expressed in the nucleus in the Vehicle group. It has been known that NF-κB is located in the cytoplasm with the inhibitory protein IκBα. After the degradation of IκB, NF-κB is activated and translocated into the nucleus where it binds to DNA [23,24,25]. Our results in the present study suggest that the PPARα agonist inhibited the translocation of NF-κB into the nucleus. During re-epithelialization, however, the IκB-α-positive labeling of the cytoplasm of the basal cells in the PPARα agonist group appeared stronger than that in the Vehicle group. This observation suggests that IκB-α was maintained in the cytoplasm with NF-κB expression in the PPARα agonist group.

In the last step of our experiment, we investigated the effect of the PPARα agonist on MCP-1. In the cornea, MCP-1 is expressed by epithelial cells and keratocytes [26]. Furthermore, MCP-1 is upregulated after an epithelial scrape injury [27]. In addition, introduction of MCP-1 into the rabbit cornea with a corneal micropocket assay leads to increased macrophage infiltration and angiogenesis [28]. Macrophages extravasate from the limbal vessels, infiltrate the stroma from superficial to deeper layers, and then migrate towards the corneal center in response to chemoattractants, such as MCP-1 [29,30]. In our present study, we showed that the MCP-1-positive areas appeared weaker in the PPARα agonist group compared to the Vehicle group, which suggests that decreased infiltration of macrophages caused by the PPARα agonist was associated with suppression of MCP-1 production.

In the present study, we clarified the expression of PPARs in normal and alkali-burned rat corneas. An ophthalmic solution of a PPARα agonist inhibited inflammation starting from the early phase after alkali injury. The observed anti-inflammatory effects of the PPARα agonist included the prevention of nuclear translocation of NF-κB and the decreased infiltration of inflammatory cells such as macrophages via suppression of MCP-1. These results suggest the possibility that an ophthalmic solution of a PPARα agonist may be a promising treatment for corneal injury.

## 4. Materials and Methods

### 4.1. Animal Model of Corneal Alkali Burn

All procedures and animal experiments were performed in compliance with the guidelines set forth by the Experimental Animal Ethics Review Committee of Nippon Medical School, Tokyo, Japan (approval number 28-012) and the Association for Research in Vision and Ophthalmology (ARVO) Statement for the Use of Animals in Ophthalmic and Visual Research.

All experiments in the present study used 8-week-old male Wistar rats (Sankyo Laboratory Service, Tokyo, Japan). After placing the animals under general isoflurane anesthesia, a corneal alkali burn was created by placing a 3.2 mm diameter circular piece of filter paper soaked in 1 N NaOH on the central cornea for 1 min. The cornea was immediately rinsed with 40 mL physiologic saline after the alkali exposure. This procedure was performed unilaterally in each rat.

### 4.2. Ophthalmic Solution of PPARα Agonist

Two kinds of ophthalmic solutions, a vehicle solution and a 0.05% fenofibrate hydrochloride solution, were compounded and used to compare the effect of the PPARα agonist. Ophthalmic vehicle solution was prepared using 0.1 mL polyoxyethylene sorbitan monooleate (Wako Pure Chemical Industries, Osaka, Japan) and 100 mL NaCl-based PBS (0.01 M; pH 7.4), which was prepared with disodium hydrogen phosphate dodecahydrate (232 g), sodium dihydrogen phosphate dihydrate (23.7 g), and distilled water (4000 mL). To prepare the 0.05% fenofibrate hydrochloride ophthalmic solution, we added 10 mg fenofibrate (Wako Pure Chemical Industries) to 20 mL vehicle solution. Fenofibrate is poorly soluble in aqueous solutions, therefore, fenofibrate hydrochloride ophthalmic solution was stirred for 70 min. All ophthalmic solutions were recognized pH 3.7 to pH 3.75 and stored at 4 °C in a refrigerator, and used within a month of the initial compounding without sterile filtration.

### 4.3. Treatment with the Ophthalmic Solutions

Ophthalmic solutions were not used for comparisons of PPAR localization. To study the role of PPARα, 0.05% fenofibrate hydrochloride ophthalmic solution (the PPARα agonist group) or vehicle (the Vehicle group) was topically instilled onto the rat’s ocular surfaces immediately after the alkali injury (*n* = 10 per time point). Topical administration was performed twice a day until the endpoint. The study did not use any of the contralateral eyes. At the endpoint (6 h, day 1, day 2, or day 7), after macroscopic examination, all rats were euthanized by exsanguination under general isoflurane anesthesia.

### 4.4. Histological and Immunohistochemical Analyses

For light microscopy (Biological Microscope, BX51 Olympus Co., Tokyo, Japan) analysis, after fixing the eyeballs in 10% buffered formalin, samples were embedded in paraffin. To compare the expression and localization of the PPAR subtypes, immunostaining was performed in both the normal cornea and the alkali-burned cornea (at 6 h, day 1, or day 7) without the use of ophthalmic solution. Histological and immunohistochemical analyses were then performed with the alkali-burned eyes (at 6 h, day 1, and day 2) to compare the effects of the PPARα agonist. Naphthol AS-D chloroacetate esterase staining was performed to detect infiltrating neutrophils. All primary antibodies used for the immunohistochemical analysis were selected based on their ability to cross-react with rat tissues. The antibodies used in the present study included: (1) monoclonal mouse anti-rat ED1 antibody (BMA, Nagoya, Japan), which was used to detect infiltrating macrophages; (2) polyclonal rabbit anti-rat PPARα antibody (Thermo Scientific, Pierce Biotechnology, Rockford, IL, USA), which was used to detect PPARα-expressing cells; (3) polyclonal rabbit anti-rat PPARβ antibody (Thermo Scientific), which was used to detect PPARβ-expressing cells; (4) monoclonal mouse anti-rat PPARγ antibody (Santa Cruz Biotechnology, Santa Cruz, CA, USA), which was used to detect PPARγ-expressing cells; (5) polyclonal rabbit anti-rat NF-κB p65 antibody (Thermo Scientific); (6) monoclonal mouse anti-rat IκB-α antibody (Santa Cruz Biotechnology, Dallas, TX, USA); and (7) monoclonal mouse anti-rat MCP-1 (MB10) antibody (IBL, Gunma, Japan), which were used to detect the major transcription factors involved in initiating proinflammatory immune responses. Immunohistochemistry for ED1, PPARα, PPARβ, PPARγ, NF-κB, IκB-α, and MCP-1 was performed with 10% buffered, formalin-fixed, paraffin-embedded tissue sections. As a control, the immunohistochemistry experiments were performed without the primary antibody or with irrelevant primary antibodies. Specimen staining was performed using the standard avidin–biotin–peroxidase complex technique.

### 4.5. Real-Time RT–PCR

For real-time RT–PCR, corneal tissues were removed from the enucleated eyes, immediately placed into RNAlater solution (Life Technologies, Carlsbad, CA, USA), and stored at −80 °C until analysis. To compare the effects of the PPARα agonist, we used real-time RT–PCR to examine the mRNA expression levels of IL-1β, IL-6, IL-8 (CXCL8), MCP-1/CCL2, VEGF-A, PPARα, NF-κB, and IκB-α. The Qiagen RNeasy Mini Kit (Qiagen, Hilden, Germany) was used to extract total RNA from corneas, and the NanoDrop ND-1000 V 3.2.1 Spectrophotometer (NanoDrop Technologies, Wilmington, DE, USA) was used to measure the RNA concentration and purity (A260/A280), in accordance with the manufacturers’ protocols. The purified total RNA had an A260/A280 ratio of 1.9–2.2. The High Capacity cDNA Reverse Transcription Kit (Applied Biosystems, Foster, CA, USA) was used to create cDNA libraries from 4 μg total RNA, in accordance with the manufacturer’s protocol. To determine the gene expression levels, we analyzed 0.3 μL cDNA with real-time quantitative RT–PCR using the THUNDERBIRD SYBR qPCR Mix (Toyobo, Osaka, Japan) based on real-time detection of accumulated fluorescence, in accordance with the manufacturer’s instructions (ABI PRISM 7900HT, Applied Biosystems). To calculate the normalized value of the mRNA expression in each sample, the relative quantity of the relevant primers was divided by the quantity of the housekeeping gene, β-actin. Sequences of the real-time RT–PCR primers used in the analyses are listed in Table 1. The SDS 2.3 software program (Applied Biosystems, Foster, CA, USA) was used for all quantifications.

### 4.6. Statistical Analyses

All results are expressed as means ± standard error. The unpaired Student’s *t*-test was performed with an analytical software program (Excel, Microsoft, Redmond, WA, USA) to evaluate the differences.

## 5. Conclusions

In the present study, we demonstrated that PPARα agonist ophthalmic solution suppresses inflammation after corneal alkali burn by suppressing proinflammatory cytokines and MCP-1, and the prevention of nuclear translocation of NF-κB in injured cornea. PPARα agonist ophthalmic solution may be a promising treatment for corneal injury.

## Reference

## Figures and Tables

**Figure 1 molecules-24-00114-f001:**
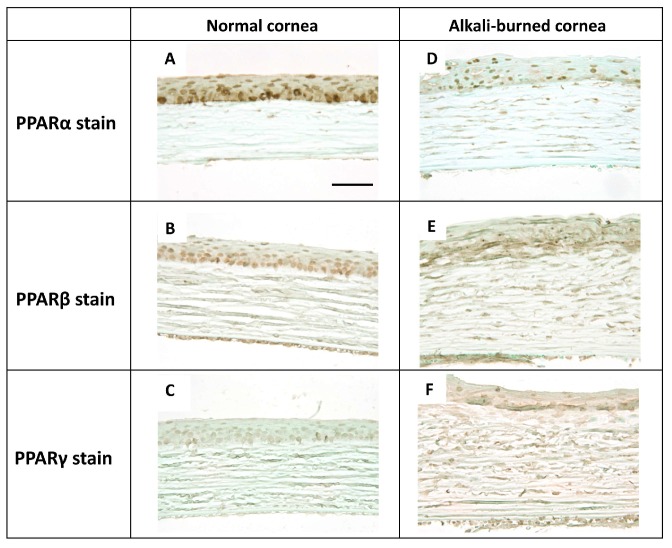
The expression of peroxisome proliferator-activated receptors (PPARs) in rat normal cornea and alkali-burned cornea at day 7. PPARα-, β-, and γ-positive cells were observed in the epithelial basement cells of normal cornea (center). (**A**–**C**; **A**: PPARα stain, **B**: PPARβ stain, **C**: PPARγ stain). In the alkali-burned cornea at day 7, PPARα, β, and γ was expressed on the re-epithelialized basement cells (**D**–**F**; **D**: PPARα stain, **E**: PPARβ stain, **F**: PPARγ stain). Scale bar: 40 μm.

**Figure 2 molecules-24-00114-f002:**
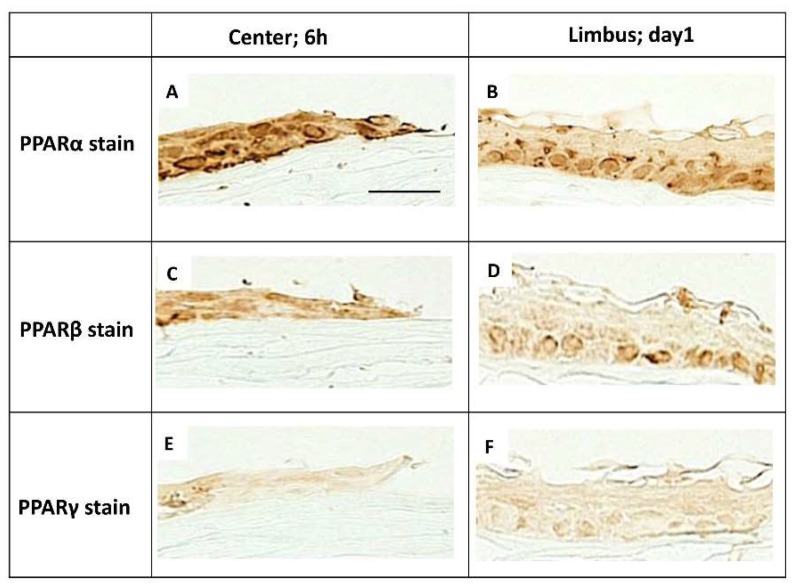
The expression of PPARs in epithelial cells in alkali-burned cornea. PPARα, β, and γ was expressed on re-epithelialized basement cells. In the epithelial basement cells of center at 6 h and limbus at day 1, the expression of PPARα (**A**,**B**) seemed to be more prominent than the expression of PPARβ (**C**,**D**) or PPARγ (**E**,**F**). Scale bar: 20 μm.

**Figure 3 molecules-24-00114-f003:**
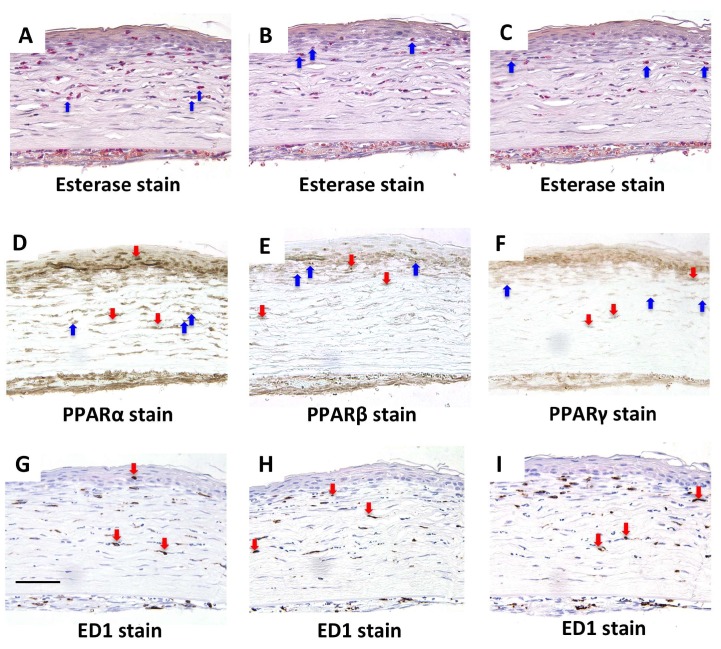
The expression and distribution of PPARs in rat cornea and infiltration of neutrophils and macrophages in alkali-burned cornea at day 7. In the alkali-burned cornea at day 7, serial sections stained with naphthol AS-D chloroacetate esterase (**A**–**C**), PPARα (**D**), PPARβ (**E**), PPARγ (**F**), and ED1 (**G**–**I**) showed that PPARs was expressed on infiltrating naphthol AS-D chloroacetate esterase positive-neutrophils (blue arrows) and ED1-positive macrophages (red arrows). Scale bar: 50 μm.

**Figure 4 molecules-24-00114-f004:**
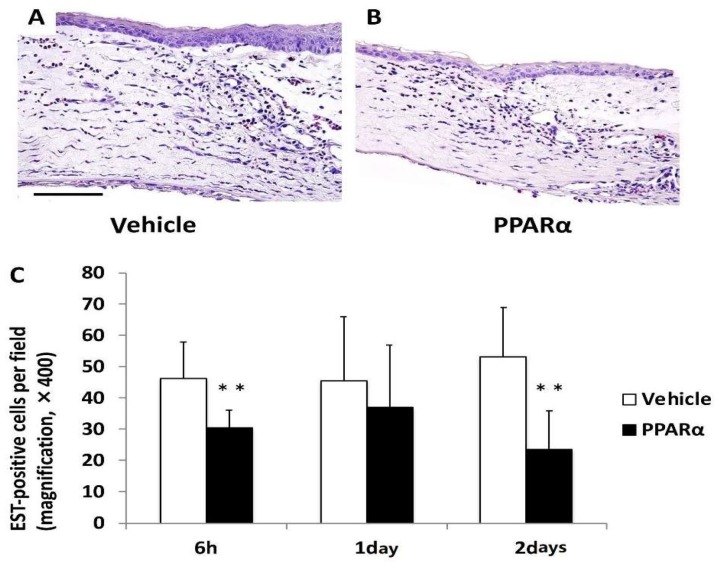
Infiltration of neutrophils in alkali-burned cornea. The number of infiltrating naphthol AS-D chloroacetate esterase (EST)-positive neutrophils (**C**) was significantly lower in the PPARα agonist group (**B**) than the vehicle group (**A**) at 6 h and on day 2 (** *p* < 0.01). Scale bar: 50 μm.

**Figure 5 molecules-24-00114-f005:**
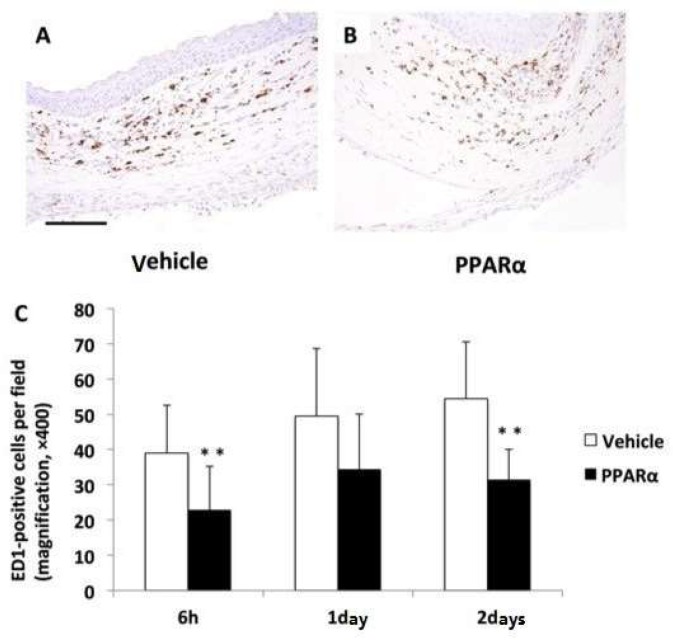
Infiltration of macrophages in alkali-burned cornea. The number of infiltrating ED1-positive macrophages (**C**) was significantly lower in the PPARα agonist group (**B**) than the vehicle group (**A**) at 6 h and on day 2 (** *p* < 0.01). Scale bar: 50 μm.

**Figure 6 molecules-24-00114-f006:**
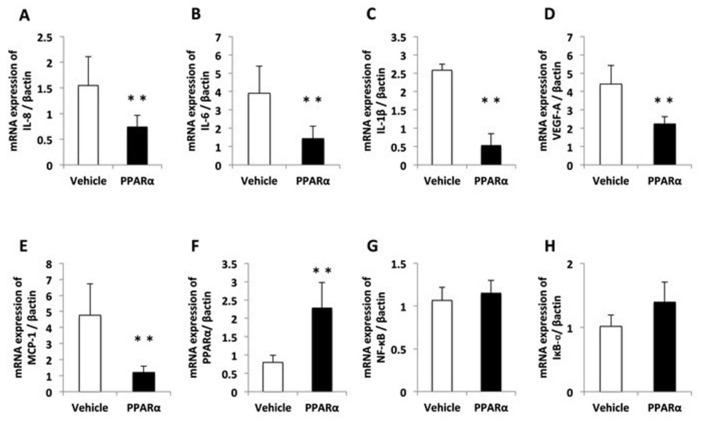
The expression of pro-inflammatory cytokines, chemokines, NF-κB, and IκB-α in the cornea after alkali burn injury. Quantification of the messenger ribonucleic acid expression levels of (**A**) IL-8 (CXCL8), (**B**) IL-6, (**C**) IL-1β, (**D**) VEGF-A, (**E**) MCP-1, (**F**) PPARα, (**G**) NF-κB, and (**H**) IκB-α. The messenger ribonucleic acid (mRNA) expression levels were measured with real-time reverse transcription–polymerase chain reaction (RT–PCR), and were normalized to the level of β-actin. A significant difference was observed between the vehicle and PPARα groups at 6 h (**A**–**F**) after injury. The levels of NF-κB and IκB-α were no significant difference (**G**,**H**). The results are presented as the means ± standard errors. ** *p* < 0.01, compared with the vehicle group.

**Figure 7 molecules-24-00114-f007:**
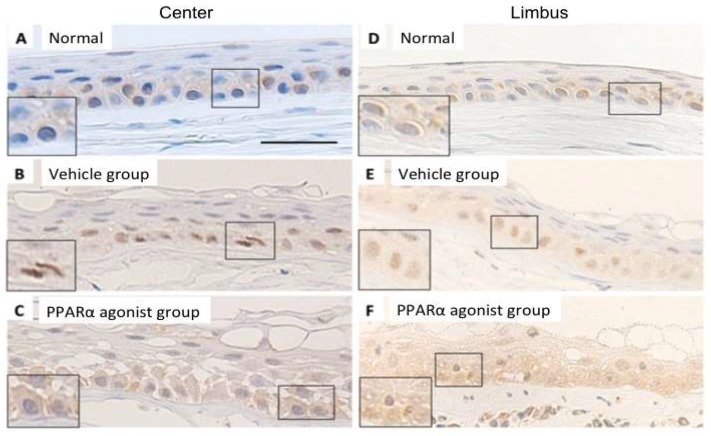
The localization of NF-κB p65 in normal cornea and in alkali-burned cornea in the vehicle and PPARα agonist groups at day 1. Immunohistochemical localization of NF-κB in normal cornea of center (**A**) and limbus (**D**) showed the weaker diffuse labeling in the cytoplasm in the epithelial basement cells. In the alkali-burned cornea of center (**B**) and limbus (**E**) in the Vehicle group, NF-κB-positive nuclei were prominent in the epithelial basement cells, indicating nuclear translocation of NF-κB developed in the alkali-burned cornea. However, in the alkali-burned cornea of center (**C**) and limbus (**F**) in the PPARα agonist groups, the location of NF-κB was noted in the cytoplasm of the epithelial basement cells, suggesting the prevention of nuclear translocation of NF-κB by PPARα agonist. Scale bar: 20 μm.

**Figure 8 molecules-24-00114-f008:**
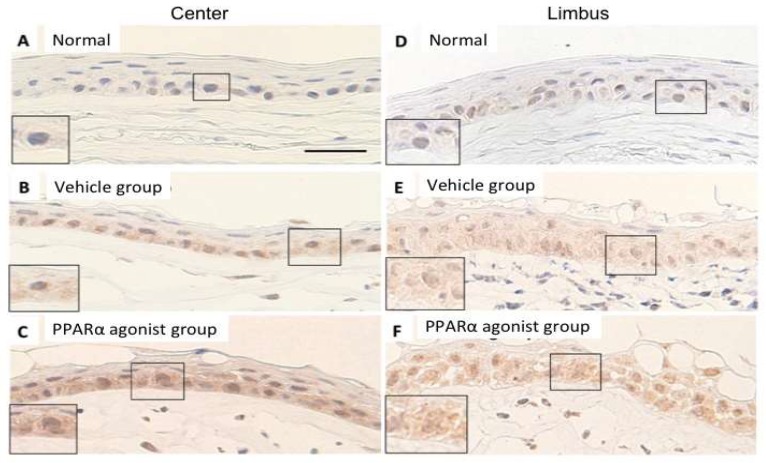
The localization of IκB-α in normal cornea and in alkali-burned cornea in the vehicle and PPARα agonist groups at day 1. Immunohistochemical localization of IκB-α in normal cornea of center (**A**) and limbus (**D**) showed the only a weak labeling in the cytoplasm in the epithelial basement cells. In the alkali-burned cornea of center (**B**,**C**) and limbus (**E**,**F**) in the Vehicle (**B**,**E**) and PPARα agonist (**C**,**F**) groups, diffuse labeling of the cytoplasm of the epithelial basement cells was observed during re-epithelialization. However, the IκB-α-positive labeling was stronger in the PPARα agonist group (**C**,**F**) compared to these regions of the Vehicle group (**B**,**E**), suggesting that PPARα agonist up-regulated IκB-α in the epithelial basement cells. Scale bar: 20 μm.

**Figure 9 molecules-24-00114-f009:**
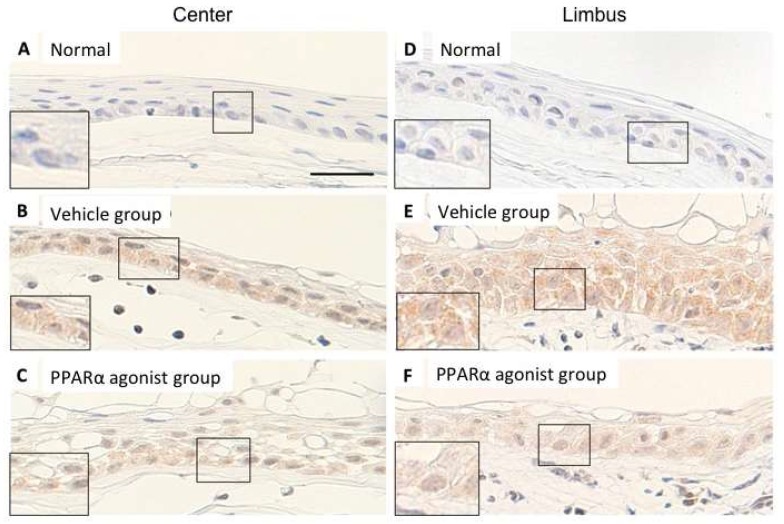
The expression of MCP-1 in rat normal cornea and in alkali-burned cornea in the vehicle and PPARα agonist groups at day 1. The expression of MCP-1 could not be detected in the epithelial basement cells in normal cornea of center (**A**) and of limbus (**D**). In the alkali-burned cornea of center (**B**,**C**) and limbus (**E**,**F**) in the Vehicle (**B**,**E**) and PPARα agonist (**C**,**F**) groups, diffuse expression of MCP-1 in the epithelial basement cells was observed during re-epithelialization. However, the expression of MCP-1 was weaker in the PPARα agonist group (**C**,**F**) compared to these regions of the Vehicle group (**B**,**E**), suggesting that PPARα agonist inhibited the expression of MCP-1 in the epithelial basement cells. Scale bar: 20 μm.

**Table 1 molecules-24-00114-t001:** Primers used for real-time RT–PCR.

Gene	Forward Primer Sequence (5′-3′)	Reverse Primer Sequence (5′-3′)
*IL*-*8 (CXCL8)*	CCCCCATGGTTCAGAAGATTG	TTGTCAGAAGCCAGCGTTCAC
*IL*-*6*	GTCAACTCCATCTGCCCTTCAG	GGCAGTGGCTGTCAACAACAT
*IL*-*1β*	TACCTATGTCTTGCCCGTGGAG	ATCATCCCACGAGTCACAGAGG
*VEGF*-*A*	TGTGCGGGCTGCTGCAATGAT	TGTGCTGGCTTTGGTGAGGTTTGA
*MCP*-*1 (CCL2)*	AGCCAGATGCAGTTAATGCCC	ACACCTGCTGCTGGTGATTCTC
*PPARα*	TGAACAAAGACGGGATG	TCAAACTTGGGTTCCATGAT
*NF*-*κB*	GGCAGCACTCCTTATCAA	GGTGTCGTCCCATCGTAG
*IκB*-*α*	TGACCATGGAAGTGATTGGTCAG	GATCACAGCCAAGTGGAGTGGA
*β*-*actin*	ACCACCATGTACCCAGGCATT	CCACACAGAGTACTTGCGCTCA

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
