# Peer review of "PPARα Agonist Suppresses Inflammation after Corneal Alkali Burn by Suppressing Proinflammatory Cytokines, MCP-1, and Nuclear Translocation of NF-κB"

_molecules, 2018, doi:10.3390/molecules24010114_

Round 1

Reviewer 1 Report

This work tries to prove the effect of a peroxisome proliferator-activated receptor agonist (PPARα) after alkaline corneal injury in rats by exploring the effect of an ophthalmic solution of the selective agonist PPARα, fenofibrate. A wide range of current and scientifically valid methodologies have been tested.  It’s very well written and supported overall by recent bibliographic references.

Comments or suggestions:

The title is too long and complex, which makes interpretation and understanding difficult.

The abstract summarises the content of the article. It’s brief and clear.

The keywords repeat the words of the title. Avoid using the same title words as this will increase the probability that the article will be detected in a search. So, I suggest that authors correct the keywords.

Introduction highlights all the subjects needed to understand all the manuscript. It's clear and detailed.

Results are very well presented.

Discussion

Line 197: “In addition, in our current study, the PPARα agonist, fenofibrate, decreased the expression of VEGF-A, which may be associated with suppressing neovascularization during the early phase after an alkali injury. Furthermore, this PPARα agonist also increased the expression of PPARα, which suggests self-inducing effects.” Authors should clarify these statements with support for relevant bibliographic references.

Line 207:” In the present study, our real-time RT–PCR analysis did not identify significantly different levels of NF-κB or IκB-α between the PPARα agonist and Vehicle groups. However, our immunohistochemical analysis demonstrated the effects of the PPARα agonist on NFκB expression. NF-κB was expressed in the cytoplasm of the basal cells in the PPARα agonist group, whereas this protein was expressed in the nucleus in the Vehicle group. This observation suggests that the PPARα agonist inhibited the translocation of NF-κB into the nucleus. During re epithelialization, however, the IκB-α-positive labeling of the cytoplasm of the basal cells in the PPARα agonist group appeared stronger than that in the Vehicle group.” Those results must be further explained.

Materials and Methods are very well described.

Author Response

We thank very much for your important comments, and believe that the manuscript has significantly improved by your suggestions. The following is a summary of the changes and our responses to the comments. Now we hope that our manuscript is good to be published.

Reviewer 1

This work tries to prove the effect of a peroxisome proliferator-activated receptor agonist (PPARα) after alkaline corneal injury in rats by exploring the effect of an ophthalmic solution of the selective agonist PPARα, fenofibrate. A wide range of current and scientifically valid methodologies have been tested. It’s very well written and supported overall by recent bibliographic references. 

Comments or suggestions: 

1. The title is too long and complex, which makes interpretation and understanding difficult. 

Response: In accordance with the comment, we have changed the title as “PPARα agonist suppresses inflammation after corneal alkali burn by suppressing proinflammatory cytokines, MCP-1, and nuclear translocation of NF-κB”.

2. The abstract summarizes the content of the article. It’s brief and clear. 

Response: Thank you very much for your comment.

3. The keywords repeat the words of the title. Avoid using the same title words as this will increase the probability that the article will be detected in a search. So, I suggest that authors correct the keywords. 

Response: In accordance with the comment, we have changed the keywords as “fenofibrate, IκB-α, IL-1β, ophthalmic solution, VEGF-A”.

4. Introduction highlights all the subjects needed to understand all the manuscript. It's clear and detailed.  

Response: Thank you very much for your comment.

5. Line 197: “In addition, in our current study, the PPARα agonist, fenofibrate, decreased the expression of VEGF-A, which may be associated with suppressing neovascularization during the early phase after an alkali injury. Furthermore, this PPARα agonist also increased the expression of PPARα, which suggests self-inducing effects.” Authors should clarify these statements with support for relevant bibliographic references. 

Response: In accordance with the comment, we have changed the sentences and add the references. “In addition, in our current study, the PPARα agonist, fenofibrate, decreased the expression of VEGF-A, which may be associated with suppressing neovascularization during the early phase after an alkali injury. The results were in good accordance with previous studies Furthermore, immunohistochemical and real-time RT–PCR analyses demonstrated that fenofibrate also increased the expression of PPARα, which suggests self-inducing effects.” (line 268 to 274 in revised manuscript)

6. Line 207:” In the present study, our real-time RT–PCR analysis did not identify significantly different levels of NF-κB or IκB-α between the PPARα agonist and Vehicle groups. However, our immunohistochemical analysis demonstrated the effects of the PPARα agonist on NFκB expression. NF-κB was expressed in the cytoplasm of the basal cells in the PPARα agonist group, whereas this protein was expressed in the nucleus in the Vehicle group. This observation suggests that the PPARα agonist inhibited the translocation of NF-κB into the nucleus. During re epithelialization, however, the IκB-α-positive labeling of the cytoplasm of the basal cells in the PPARα agonist group appeared stronger than that in the Vehicle group.” Those results must be further explained. 

Response: Thank you very much for your comments. Following the reviewer’s important comment, we have added the explanation that “It has been known that NF-κB is located in the cytoplasm with the inhibitory protein IκBα. After the degradation of IκB, NF-kB is activated and translocated into the nucleus where it binds to DNA [23-25]” in Discussion (line 290 to line 293). We have also added the explanation that “This observation suggests that IκB-α was maintained in the cytoplasm with NK-kB expression in the PPARα agonist group” in Discussion (line 296 to 297).

Reviewer 2 Report

Manuscript by Yuichiro Nakano and co-Authors is focused on  the role of PPAR alpha in the physiology of cornea. Topical application of fenofibrate in ophthalmic solution on the corneas after alkali burns significantly reduces inflammatory response, including  neutrophil and macrophage infiltration and expression levels of numerous pro-inflammatory cytokines. The possible mechanism of these effects (inhibition of the NFkappaB nuclear translocation) is presented.  The Authors conclude that the PPAR alpha agonist fenofibrate might be a successfully applied in the treatment of corneal injury after alkali burns. The study is well designed and presented, conclusions built on sound data. The chosen methods are adequate. I have only minor comments and suggestions:

1.       In the Introduction there should be a brief explanation of why inflammation is generally so dangerous in eye injuries. The rationale of anti-inflammatory therapy for the treatment of eye injuries should be explained for non experts.

2.       In the lines 101 – 109 a lot of numerical data are listed. I would suggest placing them in the table (e.g. neutrophils and macrophages in the rows, time points and groups in columns, or somehow).

3.       In the line 119 the observation is presented that the PPAR alpha level is up-regulated in the group treated with the PPAR alpha agonist. Does is mean that PPAR alpha enhances its own expression? Please comment on this in the text (in the results or discussion section).

4.        The Authors should consider and discuss the possibility that fenofibrate acts in a PPAR alpha-independent fashion. Have other PPARalpha agonists been tested in the context of ocular inflammation and show similar activity as fenofibrate? I know that generally PPAR alpha agonists have strong anti-inflammatory activities, but I found one paper about Wy 14 643, that actually exerts pro-inflammatory effect: Zhang JZ, Ward KW 2010 Int J Toxicol 29:496-504 (PMID: 20884859).

5.       In the paragraph 4.2 I was not able to find information if the ophthalmic solution was sterile filtered or sterilized in any other way, prior to use. Please clarify. Fenofibrate is very poorly soluble in aqueous solutions. I would appreciate any information about fenofibrate solubility in this experimetal setting. Did the Authors encountered any problems with it?

6.       I know that the manuscript template for Molecules Journal suggests the manuscript structure as: Introduction, Results, Discussion, Materials and Methods and Conclusions, but for me it seems awkward to have between Discussion and Conslusion sections separated by Materials and Methods in between. I would prefer to move M&M at the end.

Author Response

We thank very much for your important comments, and believe that the manuscript has significantly improved by your suggestions. The following is a summary of the changes and our responses to the comments. Now we hope that our manuscript is good to be published.

Reviewer 2

Manuscript by Yuichiro Nakano and co-Authors is focused on the role of PPAR alpha in the physiology of cornea. Topical application of fenofibrate in ophthalmic solution on the corneas after alkali burns significantly reduces inflammatory response, including neutrophil and macrophage infiltration and expression levels of numerous pro-inflammatory cytokines. The possible mechanism of these effects (inhibition of the NFkappaB nuclear translocation) is presented. The Authors conclude that the PPAR alpha agonist fenofibrate might be a successfully applied in the treatment of corneal injury after alkali burns. The study is well designed and presented, conclusions built on sound data. The chosen methods are adequate. I have only minor comments and suggestions:

Comments or suggestions:

1. In the Introduction there should be a brief explanation of why inflammation is generally so dangerous in eye injuries. The rationale of anti-inflammatory therapy for the treatment of eye injuries should be explained for non experts.

Response:In accordance with the comment, we have added the sentence “For good vision, transparency of the cornea is essential. Injuries that damage the corneal stroma often result in scarring. Since inflammation and angiogenesis are deeply involved in scar tissue formation, agents that can suppress these phenomena have long been sought. One of the new suppression candidates is peroxisome proliferator-activated receptors (PPARs).” in the introduction. (line 61 to 65 in Introduction)

2. In the lines 101 – 109 a lot of numerical data are listed. I would suggest placing them in the table (e.g. neutrophils and macrophages in the rows, time points and groups in columns, or somehow).

Response:In accordance with the comment, Figures 1 and 2 were modified into the table format to facilitate understanding of the data.

3. In the line 119 the observation is presented that the PPAR alpha level is up-regulated in the group treated with the PPAR alpha agonist. Does is mean that PPAR alpha enhances its own expression? Please comment on this in the text (in the results or discussion section).

Response:PPARα enhanced its own expression. Comments in Discussion have been revised as line 272-274. We changed the comment, “Furthermore, immunohistochemical and real-time RT–PCR analyses demonstrated that fenofibrate also increased the expression of PPARα, which suggests self-inducing effects.”

4. The Authors should consider and discuss the possibility that fenofibrate acts in a PPAR alpha-independent fashion. Have other PPARalpha agonists been tested in the context of ocular inflammation and show similar activity as fenofibrate? I know that generally PPAR alpha agonists have strong anti- inflammatory activities, but I found one paper about Wy 14 643, that actually exerts pro-inflammatory effect: Zhang JZ, Ward KW 2010 Int J Toxicol 29:496- 504 (PMID: 20884859).

Response:As the reviewer commented, PPARα WY 14643 has been reported to have pro-inflammatory effect or anti-inflammatory effect depending the situation. The detail explanation of the difference has yet to be available. Thus, in the present study we expected its anti-inflammatory effect in our corneal injury model and the effect was confirmed. We do not have any evidence that the anti-inflammatory phenomenon found in the present study was induced by PPARα independent fashion, which strongly suggested the anti-inflammatory effect of PPARα itself.

5. In the paragraph 4.2 I was not able to find information if the ophthalmic solution was sterile filtered or sterilized in any other way, prior to use. Please clarify. Fenofibrate is very poorly soluble in aqueous solutions. I would appreciate any information about fenofibrate solubility in this experimental setting. Did the Authors encountered any problems with it?

Response:In accordance with the comment, we have added the sentences “fenofibrate is poorly soluble in aqueous solutions, therefore, fenofibrate hydrochloride ophthalmic solution was stirred for 70 minutes in the present study. All ophthalmic solutions were recognized pH 3.7 to pH3.75 and stored at 4°C in a refrigerator, and used within a month of the initial compounding without sterile filtration.” in 4. Materials and methods (line 335 to 339 in revised manuscript).

6. I know that the manuscript template for Molecules Journal suggests the manuscript structure as: Introduction, Results, Discussion, Materials and Methods and Conclusions, but for me it seems awkward to have between Discussion and Conclusion sections separated by Materials and Methods in between. I would prefer to move M&M at the end.

Response:Thank you for suggestion. As you mentioned that the manuscript template for Molecules Journal suggests the manuscript structure as: Introduction, Results, Discussion, Materials and Methods and Conclusions. Following your comment, we move our conclusion to end of discussion (line 310 to 317 in revised manuscript), and added the brief description “In the present study, we demonstrated that PPARα agonist ophthalmic solution suppresses inflammation after corneal alkali burn by suppressing proinflammatory cytokines and MCP-1, and the prevention of nuclear translocation of NF-κB in injured cornea. PPARα agonist ophthalmic solution may be a promising treatment for corneal injury.” in Conclusion after M&M (line 417 to 421 in revised manuscript).

Reviewer 3 Report

The effect of a peroxisome proliferator-activated receptor α (PPARα) agonist (Fenofibrate) after corneal alkali injury was studied. The number of infiltrating neutrophils and macrophages at the corneal limbus was lower in the PPARα agonist group. Interleukin-1β (IL-1β), IL-6, IL-8, monocyte chemoattractant protein-1 (MCP-1), and vascular endothelial growth factor-A mRNA expression was suppressed in the PPARα agonist group. NF-κB was expressed in the cytoplasm of basal cells in the PPARα agonist group and in the nucleus in the Vehicle group. MCP-1 was more weakly expressed in the PPARα agonist group.

This study was interesting and well-design, but some points must be clarified.

Three type of PPAR including PPARα, β/δ, and γ were study in this work. The role of different PPAR play in inflammation after corneal alkali burn must state and discuss clearly.

Several factors related to inflammation were studied in this work. The regulation between the factors has to descript clearly.

Some photos in this work were unsatisfactory, especially Figure 3. Please replace with clear ones.

Author Response

We thank very much for your important comments, and believe that the manuscript has significantly improved by your suggestions. The following is a summary of the changes and our responses to the comments. Now we hope that our manuscript is good to be published.

Reviewer 3

The effect of a peroxisome proliferator-activated receptor α (PPARα) agonist (Fenofibrate) after corneal alkali injury was studied. The number of infiltrating neutrophils and macrophages at the corneal limbus was lower in the PPARα agonist group. Interleukin-1β (IL-1β), IL-6, IL-8, monocyte chemoattractant protein-1 (MCP-1), and vascular endothelial growth factor-A mRNA expression was suppressed in the PPARα agonist group. NF-κB was expressed in the cytoplasm of basal cells in the PPARα agonist group and in the nucleus in the Vehicle group. MCP-1 was more weakly expressed in the PPARα agonist group. This study was interesting and well-design, but some points must be clarified.

Comments or suggestions:

1. Three type of PPAR including PPARα, β/δ, and γ were study in this work. The role of different PPAR play in inflammation after corneal alkali burn must state and discuss clearly.

Response:In accordance with the comment, we have added the sentence, “It has been reported that PPARα, β/δ, and γ plays different roles in corneal wound healing. Administration of PPARα not only reduces inflammation but also corneal neovascularization [12]. PPARβ is involved in tissue repair and administration of PPARβ agonist promotes the healing of corneal epithelial wounds [10]. The ophthalmic solution of the PPARγ agonist inhibits inflammation and furthermore promotes corneal wound healing by activating M2 macrophage [11].” In Discussion. (line 246-252.)

2. Several factors related to inflammation were studied in this work. The regulation between the factors has to descript clearly.

Response:In accordance with the comment, we have added the sentence, “Pro-inflammatory cytokines, IL-1β, IL-6, IL-8, chemokine MCP-1, and VEGF are deeply involved in inflammation. In the process, activated NF-kB is known to be associated with production of pro-inflammatory cytokines, IL1β, IL6, and IL8.” in line 278-281. 

3. Some photos in this work were unsatisfactory, especially Figure 3. Please replace with clear ones.

Response:In accordance with the comment, we have changed the Figure 3.
